# Evolution and development of fruits of *Erycina pusilla* and other orchid species

Dewi Pramanik[1,2,3]*, Annette Becker[4], Clemens Roessner[4], Oliver Rupp[5], Diego Bogarín[1,6], Oscar Alejandro Pérez-Escobar[7], Anita Dirks-Mulder[8], Kevin Droppert[8], Alexander Kocyan[9], Erik Smets[1,2,10], Barbara Gravendeel[1,2,11]

1 Evolutionary Ecology Group, Naturalis Biodiversity Center, Leiden, The Netherlands, 2 Institute of Biology Leiden, Leiden University, Leiden, The Netherlands, 3 National Research and Innovation Agency Republic of Indonesia (BRIN), Central Jakarta, Indonesia, 4 Development Biology of Plants, Institute for Botany, Justus-Liebig-University Giessen, Giessen, Germany, 5 Department of Bioinformatics and Systems Biology, Justus Liebig University, Giessen, Germany, 6 Jardín Botánico Lankester, Universidad de Costa Rica, Cartago, Costa Rica, 7 Comparative Plant and Fungal Biology, Royal Botanicals Gardens, Kew Richmond Surrey, London, United Kingdom, 8 Faculty of Science and Technology, University of Applied Sciences Leiden, Leiden, The Netherlands, 9 Botanical Museum, Department of Plant and Microbial Biology, University of Zurich, Zurich, Switzerland, 10 Ecology, Evolution and Biodiversity Conservation, KU Leuven, Heverlee, Belgium, 11 Radboud Institute for Biological and Environmental Sciences, Radboud University, Nijmegen, The Netherlands

* dewi.pramanik@naturalis.nl

**Data Availability Statement:** All nucleotides data are available from the GenBank (NCBI) database (accession number(s) are available in S4–S7 Tables).

## Abstract

Fruits play a crucial role in seed dispersal. They open along dehiscence zones. Fruit dehiscence zone formation has been intensively studied in *Arabidopsis thaliana*. However, little is known about the mechanisms and genes involved in the formation of fruit dehiscence zones in species outside the Brassicaceae. The dehiscence zone of *A. thaliana* contains a lignified layer, while dehiscence zone tissues of the emerging orchid model *Erycina pusilla* include a lipid layer. Here we present an analysis of evolution and development of fruit dehiscence zones in orchids. We performed ancestral state reconstructions across the five orchid subfamilies to study the evolution of selected fruit traits and explored dehiscence zone developmental genes using RNA-seq and qPCR. We found that erect dehiscent fruits with non-lignified dehiscence zones and a short ripening period are ancestral characters in orchids. Lignified dehiscence zones in orchid fruits evolved multiple times from non-lignified zones. Furthermore, we carried out gene expression analysis of tissues from different developmental stages of *E. pusilla* fruits. We found that fruit dehiscence genes from the MADS-box gene family and other important regulators in *E. pusilla* differed in their expression pattern from their homologs in *A. thaliana*. This suggests that the current *A. thaliana* fruit dehiscence model requires adjustment for orchids. Additionally, we discovered that homologs of *A. thaliana* genes involved in the development of carpel, gynoecium and ovules, and genes involved in lipid biosynthesis were expressed in the fruit valves of *E. pusilla*, implying that these genes may play a novel role in formation of dehiscence zone tissues in orchids. Future functional analysis of developmental regulators, lipid identification and quantification can shed more light on lipid-layer based dehiscence of orchid fruits.

**Funding:** This study was financially supported by a personal grant from the Sustainable Management for Agriculture Research and Development (SMARTD) project of Badan Penelitian dan Pengembangan Pertanian, Indonesia number 133/KPTS/Kp.320/02/2018 to Dewi Pramanik. The funders had no role in study design, data collection and analysis, publication decision, or manuscript preparation.

**Competing interests:** The authors have declared that no competing interests exist.

## Introduction

The angiosperm diversification record extends back to the early Cretaceous [1]. Since their appearance in the fossil record, angiosperms have radiated massively resulting in species richness and ecological dominance surpassing all other plant lineages [2]. They comprise approximately 350,500 described species [3]. The main evolutionary innovation of angiosperms is the ability to produce flowers [4] and gynoecia (the sum of all carpels) to protect developing ovules and seeds. The fruit is a structure that develops in flowering plants from the gynoecia after fertilization [5]. Different fruit types in angiosperms evolved with free (apocarpous) or (partly) fused carpels (syncarpous), a dry or fleshy texture, various pericarp layer differentiations, and different ways of opening [6–8]. The most abundant fruit types are dry dehiscent, dry indehiscent and fleshy, with frequent transitions between the different fruit types [9].

The gynoecium of *A. thaliana* is a silique consisting of two carpels forming one central cavity that is divided in two by a false septum or replum. After fertilization and fruit maturation, the fruit opens by separation of the two valves from the replum. The dehiscence zone (DZ) lies at the boundaries between the valves and the replum. At maturity, the DZ forms a separate layer that is not lignified, placed between lignified cells in the valve and the lignified vasculature of the replum [10]. The development and separation of the DZ are controlled by a complex molecular network, consisting of a dynamic interplay among hormones and transcription factors [11]. *FRUITFULL* (*FUL*) plays a pivotal role in valve development [10, 12], and *REPLUMLESS* (*RPL*) promotes replum development [13]. Meanwhile, *SHATTERPROOF 1/2* (*SHP1/2*) controls valve margin formation and dehiscence [14]. These genes activate other valve margin genes, e.g. *INDEHISCENT (IND)* and *ALCATRAZ (ALC)* [15, 16]. *IND* is expressed in both the lignified and separation layer [14] whereas *SPT/ALC* is expressed in the separation layer only [15]. Together with *RPL*, *FUL* represses the expression of *SHP1/2* [12, 17]. Combined action of *SHP1/2*, *IND*, and *ALC* forms a regulatory network for valve margin formation and proper specification of different cell types within the valve margin and DZ [14, 17]. In addition to these genes, the class I *KNOTTED1*-like homeobox (*KNOX*) gene *BREVIPEDICELLUS* (*BP* aka *KNAT1*) is expressed in valve margin and replum, and together with *RPL*, promotes replum formation [18]. While *APETALA2* (*AP2*), *FILAMENTOUS FLOWER* (*FIL*), *YABBY3* (*YAB3*), and *JAGGED* (*JAG*) repress *BP/RPL*, these genes are also crucial for precise positioning of the valve margin by preventing replum and valve margin overgrowth [19, 20]. Plant-specific YABBY proteins such as *CRABS CLAWS*-like/DROOPING-LEAF-like (*CRC/DL*) genes may also be involved in fruit DZ formation. *CRC/DL* genes are the key developmental regulators for carpel and gynoecium development in plants [21].

The orchid family is known for its spectacular flower shape and size diversity. Less known is that orchids also display very diverse fruit morphologies [22]. There are two main types of orchid fruits: indehiscent fleshy and dehiscent dry ones. Fleshy indehiscent orchid fruits are rare and only occur in some species of *Apostasia* and *Neuwiedia* (subfamily Apostasioideae), *Selenipedium* (subfamily Cypripedioideae), *Cyrtosia*, *Pogoniopsis*, and *Vanilla* (subfamily Vanilloideae), *Corymborkis*, *Palmorchis*, and *Yoania* (subfamily Epidendroideae) [23–27]. The prominent characteristic of fleshy indehiscent orchid fruits is the hard and thick seed coat as an adaptation to endozoochory [28, 29]. Dry dehiscent orchid fruits evolved in 99% of the species of the Orchidaceae. The seeds of dehiscent orchid fruits have fragile, thin coats formed by single-cell layers and non-lignified cells surrounding the embryo [24] as an adaptation to wind dispersal.

Since genes involved in fruit formation of orchids are unknown, we used information from *A. thaliana* as a starting point knowing about the fundamental differences between fruit development and dehiscence of orchids and *A. thaliana*. At maturity, dehiscence in *A. thaliana*

occurs at the DZ placed between a region of lignified cells in the valve and the lignified vasculature of the replum. The resulting valves detach from the replum freeing the seeds [10, 30]. In contrast to *A. thaliana*, and according to the "split carpel model" [22], inferior orchid fruits consist of six valves, three broad fertile ones, bearing the ovules, and three narrow sterile valves (Fig 1). The sterile valves develop from the sepal bases and in many cases may appear to split the carpel opposite to it. In contrast, the fertile valves each develop from two carpel-halves and

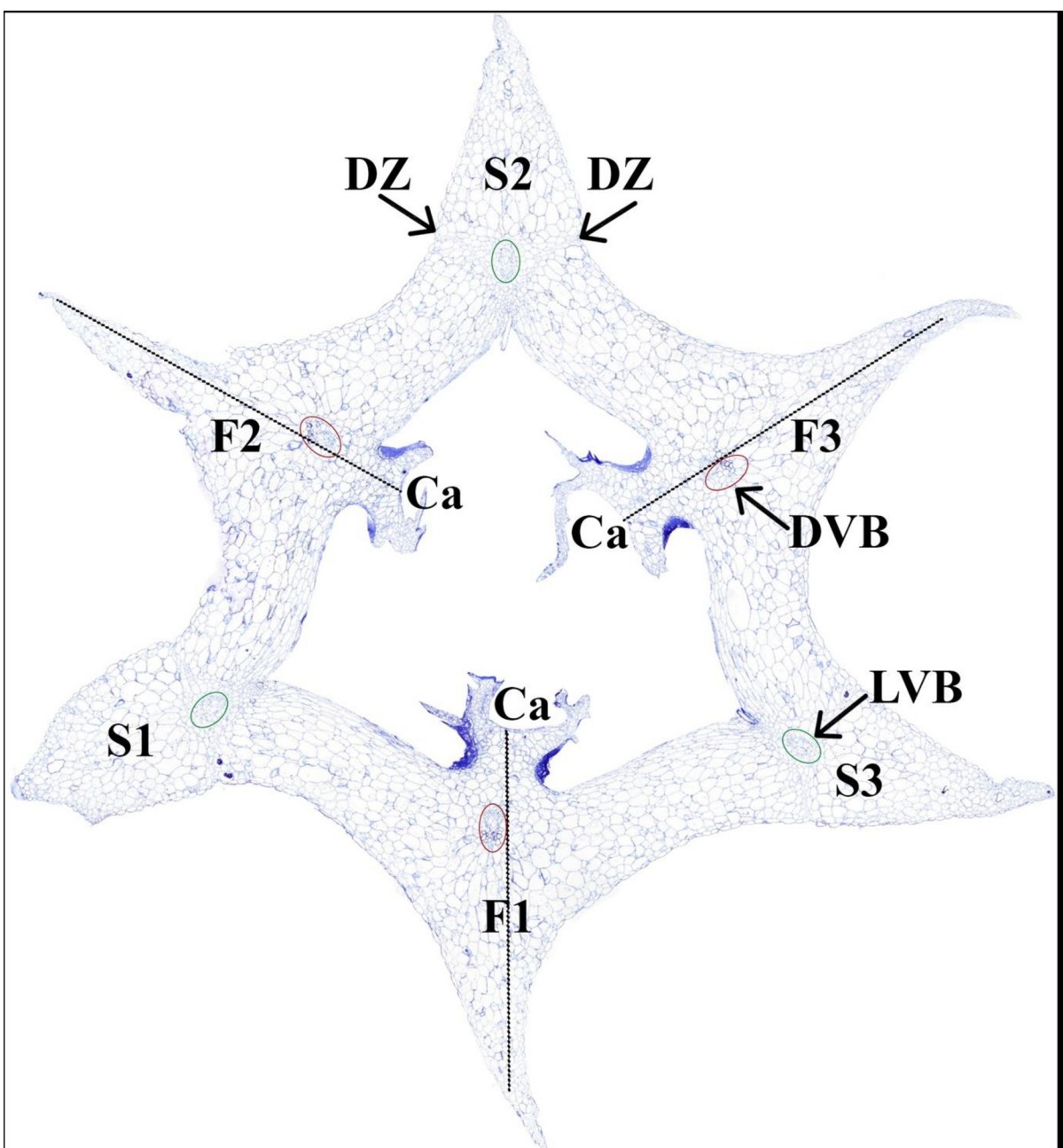

**Fig 1. Cross section of *Erycina pusilla* fruit 8 weeks after pollination reflecting the split carpel model according to Rasmussen and Johansen (2006).** Abbreviations: F1 = fertile valve originating from labellum; F2 = fertile valve originating from lateral petal 1; F2 = fertile valve originating from lateral petal 2; S1 = sterile valve originating from lateral sepal 1; S2 = sterile valve originating from dorsal sepal; S3 = sterile valve originating from lateral sepal 2; Ca = carpel; DVB = dorsal vascular bundle; LVD = lateral vascular bundle.

a petal base, carrying one marginal placenta [22] (Fig 1). When a dehiscent orchid fruit grows in size and width, pressure is slowly building up, and when the fruit is ripe it splits open in the areas between the valves [22]. The valves of the *E. pusill*a fruit begin to separate longitudinally 16 weeks after pollination (WAP), starting at the apex and remaining fused at the base. The fruit begins to open at the endocarp borders of the sterile valves and continues to open along the DZ at the boundaries of fertile and sterile valves by rupturing the cells. Typically, the fruit splits into three wide seed bearing fertile valves along three of six fusion zones, while the narrow sterile valves remain connected to one of the fertile valves [31].

*E. pusilla* fruit DZ cells appear within one WAP at the border between the sterile and fertile valve margins. These cells are fully developed at three WAP and then consist of two layers [31] (Fig 1). Even though the development of DZ's has been described, little information about genes involved in DZ development and differentiation has been obtained for orchids. Combined transcriptome and qPCR analyses revealed expression of MADS-domain transcription factors like *EpAG*, *EpSEP*, *EpAGL6*, *EpSVP* and the *A. thaliana* dehiscence-related genes *HEC3*, *RPL*, and *SPT/ALC* in ripening fruits of *E. pusilla* [31]. However, these authors did not focus on development of the DZ and genes specifically related to DZ cell formation. They discovered that a layer of cuticle lipids is formed in the DZ of *E. pusilla* fruits instead of the lignification patterning well known from *A. thaliana*. Candidate genes involved in cuticle lipid biosynthesis can be derived from triacylglycerol (TAG), wax, or cutin biosynthesis. The specific genes and their role in the formation of cuticle lipids in the DZ of *E. pusilla* are still unknown.

We aimed to study the evolution and development of DZ formation in orchid fruits in more detail. First, we examined the morphology and anatomy of fruits of 44 orchid species representing 31 genera of five subfamilies to investigate the evolutionary relationship between fruit traits by ancestral state reconstruction (ASR) analysis. Next, we studied the expression of a selection of candidate genes during DZ development of fruits of the emerging orchid model *E. pusilla*. We formulated the following hypotheses: (1) DZ lignification and shorter fruit ripening time co-evolved, especially in species coping with relatively short seasons suitable for seeds dispersal; (2) there are different expression patterns of *SEP*, *SVP*, *SPT*, *HEC3*, and *RPL* orthologs in the DZ of *E. pusilla* as compared with *A. thaliana*, as both species have a morphologically different DZ: lignified versus non lignified; and (3) expression of genes involved in lipid synthesis is upregulated in older fruits of *E. pusilla* due to the presence of a lipid layer in the DZs.

## Material and methods

### Morphological analysis

**Scoring of morphological traits.** Based on differences in morphological characteristics of orchid fruits in ripening and opening, we selected six different key morphological traits that were straightforward to score, i.e. fruit ripening period ($\leq 4$ or $> 4$ months), fruit orientation (erect or pendant), fruit dehiscence (indehiscent or dehiscent), number of slits in mature fruits ($< 3$ or $\geq 3$), fusion of valves during dehiscence (fused at base and apex, or fused at the base only), and lignification of the valves and/or DZ (either non-lignified or lignified). These characters were scored from freshly harvested ripe fruits of a selection of orchid species cultivated at the Hortus Botanicus Leiden (The Netherlands), Lankester Botanical Garden (Costa Rica), Utrecht Botanic Gardens (The Netherlands) and Bochum Botanical Garden (Germany) (S1 Table).

**Lignification of orchid fruits.** A survey of fruit DZ's was made using phloroglucinol staining of mature freshly harvested fruits. Handmade cross-sections of fruits were treated

with 100% lactic acid for an hour at 60˚C to clear the tissue. The sections were subsequently stained using 1% phloroglucinol in 96% ethanol for an hour, then immersed in 25% (v/v) hydrochloric acid for 2–5 minutes until the stain was visible and immediately examined under a Binocular microscope (Stereo Discovery V12, Carl Zeiss Microscopy GmbH., Jena, Germany). Sections were washed using demineralized water in between each step. Lignification of valve and/or DZ tissue was scored as present (1) or absent (0) (S1 and S2 Tables).

## Ancestral state reconstruction

We aligned plastid *mat*K and *rbc*L and nuclear ribosomal ITS sequences of 45 orchid species selected across the five orchid subfamilies (S3 Table), collected from NCBI GenBank (https://www.ncbi.nlm.nih.gov/genbank/) [32]. In addition, we included sequences of two accessions of *Hypoxis curtissii* Rose (Hypoxidaceae) and two of *Narcissus bulbocodium* L. (Amaryllidaceae) as outgroups due to the close relationship of these families with orchids within the monocots [33]. We aligned and trimmed the matrices of each marker in Geneious Prime® R2021.2.2 (Biomatters Ltd., Auckland, New Zealand) using MAFFT (Multiple Alignment using Fast Fourier Transform). The concatenated dataset (nrITS+*mat*K+*rbc*L) was built with concatenated sequences in Geneious Prime® R2021.2.2 (Biomatters Ltd., Auckland, New Zealand). When sequences for a specific marker were not available, they were included as missing data. When a species analysed morphologically was not available in NCBI GenBank, it was substituted in the DNA concatenated matrix by a DNA sequence of a close relative.

We estimated the divergence times in BEAST v.1.8.2 using the CIPRES Science Gateway to obtain ultrametric trees for the ancestral state reconstructions [34]. The MCMC parameters were set to $20\times10^6$ generations and a sampling frequency of 1000, yielding 20,001 trees per run. The substitution model selected was GTR, estimated with 4 gamma categories, estimated lognormal relaxed clock (uncorrelated) and the Yule Process (Y) speciation tree model. A normal prior distribution of 105.06 (±2.5 standard deviations) Ma was assigned to the root node of the Orchidaceae and 94.75 (±3.5 standard deviations) Ma to the node containing all Orchidaceae members. These secondary calibrations were obtained from various dating studies of the Orchidaceae [33, 35–37]. The convergence of independent runs and the MCMC parameters (ESS values >200) were inspected in Tracer v.1.6. Finally, a maximum clade credibility (MCC) tree was obtained with a 10% burnin using TreeAnnotator v.1.8.2. Resulting trees and the 95% highest posterior density (HPD) estimations were visualized in FigTree v1.4.3 [38] and manipulated with R programming language (R Core Team, 2018) under R Studio [39], using the packages APE, ggtree and phytools [40–42].

Ancestral state reconstructions (ASRs) were assessed with ML and stochastic character mapping (SCM) using ultrametric trees. For the ML approach, we tested several models: equal rates (ER), symmetrical (SYM) and all rates different (ARD) with the re-rooting method of Yang et al. [43] and the function ACE implemented in R [44] packages APE, ggtree and phytools [40–42]. The best-fitting models were selected by a likelihood ratio test. For the SCM analysis we performed 100 replicates on 100 randomly selected trees (10,000 mapped trees) derived from the BEAST analysis. These trees were randomly selected using the R function *samples.trees* (http://coleoguy.blogspot.de/2012/09/randomly-sampling-trees.html). The functions *make.simmap* and *describe.simmap* were applied to produce the time spent of transitions and the proportion in each state [41, 45]. The mean probabilities retrieved at each node were plotted with phytools on the MCC tree for each character analyzed.

We tested correlations among traits using the program BayesTraits V3.0.1 [46–48] by performing 81 comparisons between each pair of the 9 characters of fruits assessed. We tested two models under Bayesian Inference: a dependent model which allows correlation among traits

against the independence model with no correlation among traits (correlations set to zero). We used a set of 1000 trees (randomly selected with the R function *samples.trees* as described above) from the post burnin sample of the 20,001 ultrametric trees obtained from the time-calibrated BEAST analysis to record for phylogenetic uncertainty. The MCMC parameters of each model were set to 1,010,000 iterations, sample period of 1,000, burnin of 10,000, auto tune rate deviation and stepping stones 100 10,000 with reversible jump hyper-prior exponential. The BayesTraits outputs files were analysed in R with the BayesTraits wrapper (btw) by Randi H Griffin (http://rgriff23.github.io/projects/btw.html) and other functions from btrtools and BTprocessR (https://github.com/hferg). The MCMC stationarity of parameters (ESS values >200) and convergence of chains was checked with the R package coda [49] and the function *mcmcPlots* of BTprocessR. Using the log marginal likelihoods obtained from the outputs, we estimated a log Bayes factor (logBF) for the dependent model and the independent model with: logBF = 2*(SS dependent model (complex model)–SS independent model (simple model)). We interpreted the logBF as suggested in the BayesTraitsV3 manual (http://www.evolution.rdg.ac.uk/BayesTraitsV3.0.1/Files/BayesTraitsV3.Manual.pdf): < 2: weak evidence; > 2: positive evidence; 5–10: strong evidence and > 10: very strong evidence.

## Molecular analysis

**Tissue culture and pollination of *E. pusilla*.**   Plants of *E. pusilla* were grown *in vitro* conditions in a sterile flask in climate cabinets from seed germination, seedling, maturation, and pollination to fruit production. The plants were cultured in a climate cabinets (Model SGC120, Weiss Technik, Loughborough, UK) at 22°C, 9.00 to 21.00 h light regime, and 50% relative humidity. For seed germination, we used multiplication media (Phytamax orchid multiplication medium, P6793, Sigma Aldrich). Whereas for the rest of the stages, we used Duchefa Orchimax medium (O027, Duchefa). Both the multiplication medium and the Orchimax medium were supplemented with 20 g/l banana powder (B1304.005, Duchefa) and 2 g/l charcoal (C1302, Duchefa). The pH was adjusted to 5.7–5.8, and afterwards, 4 g/l gelrite was added (G1101.0500, Duchefa). The medium was autoclaved for 25 minutes at 121°C with 1.5 Psi.

Fruits were produced by hand-pollination. Vigorous *in vitro* plantlets with fully open flowers were selected for *in vitro* pollination. The *in vitro* pollination was done in a laminar flow cabinet by removing the anther cap and then placing two pollinia on the stigma inside the column. After pollination, plantlets were cultured in a fresh medium to avoid fungal and bacterial contaminations and to ensure sufficient nutrition for the development of fruits. Specific fruit stages, 1 WAP and 3 WAP, were harvested for RNA-seq. Meanwhile, the rest were harvested at 12–14 WAP for seed germination. It took 20 weeks for the seeds to develop into mature flowering plants.

**RNA isolation.**   *E. pusilla* fruits were freshly harvested and dissected. Both 1 WAP and 3 WAP fruits of *E. pusilla* (S1 Fig) were manually dissected in triplicates for whole transcriptome sequencing and qPCR confirmation. In addition, fruit tissues from 0 WAP, 1 WAP, 3 WAP, 7 WAP, and 13 WAP, and floral bud tissues were pooled as a positive control for qPCR.

Fruit valve tissues were collected in 2.2-mL microcentrifuge Eppendorf tubes filled with a sterile 7-mm glass bead (Assistant, NL) and immediately immersed in liquid nitrogen. Frozen samples were stored at -80°C until all samples had been collected. The samples were subsequently ground by using a TissueLyser II (Qiagen, Venlo, NL).

RNA from these tissues was extracted using the PicoPure™ RNA Isolation Kit (Applied Biosystems™, Foster City, California, USA) following the manufacturer's instructions for whole transcriptome sequencing. For qPCR, the samples were extracted using the RNeasy Plant Mini Kit (Qiagen, Venlo, NL) under RNase-free condition, following the manufacturer's

instructions. DNase I Amp Grade (Invitrogen 1U/µl) was applied to the extracted RNA to digest single- and double-stranded DNA. Quality and quantity of the RNA were measured using a Dropsense 96 SPEC-07 (Trinean NV/SA, Gentbrugge, Belgium). The yield of all samples used was higher than 50 ng/µl, whereas RNA quality was around 1.8 at 260/230 ratio and 1.8–2.2 at 260/280 ratio. RNA integrity numbers (RIN) were evaluated using an Agilent bioanalyzer (Agilent Technologies, Santa Clara, California, USA) with the Agilent RNA 6000 Pico kit (Agilent Technologies, Santa Clara, California, USA). Agarose gel electrophoresis was applied to identify RNA degradation and potential contamination.

**Library preparation and sequencing.** The NEB Next® Ultra™ RNA Library Prep Kit (New England Biolabs, Ipswich, MA, UK) was used for library preparation. Library preparation started with mRNA enrichment to increase exonic regions by polyadenylic acid (poly(A)) enrichment. The mRNA was selected using poly(A) enrichment by mixing RNA with oligo (dT) primers. Subsequently, RNA was fragmented into an appropriate size by adding a fragmentation buffer. Then cDNA was synthesized using an mRNA template and random hexamers primers. A custom second-strand synthesis buffer (Illumina Inc., San Diego, California, USA), dNTPs, RNase H and DNA polymerase I were added to initiate second-strand synthesis. After a series of terminal repairs, ligation and sequencing adapter ligation, the double-stranded cDNA library was completed through size selection and PCR enrichment. Quality control of the library was tested by Qubit 2.0 for library concentration. Agilent 2100 tested the insert size and qPCR quantified the library effective concentration precisely. Library preparation and sequencing were done at Novogene in Cambridge, United Kingdom. Twelve samples per lane were sequenced on an Illumina Novaseq6000 (Illumina Inc., San Diego, California, USA) to generate at least 30 million reads per sample. Paired-end sequencing was applied for each sample to generate a read length of approximately 150 bp.

**Bioinformatic analysis.** Raw sequencing reads of the six samples of *E. pusilla* fruit tissues were imported to perform quality tests, trimming, RNA-seq and PCA analysis using default parameters. Assembly and annotation were carried out in the GenDB genome annotation system. Assembly and annotation of the transcriptomes were done using *de novo* assemblies with Trinity [50], rnaSPAdes [51], scSPAdes [52], Bridger (k = 25), Bridger (k = 31) [53] and reference-based assemblies with GeMoMa [54], StringTie [55]/TransBorrow [56]. Individual assemblies were merged with the EviGene pipeline [57]. The trimmed reads were pseudo-mapped with salmon [58]. The evaluation of assembly completeness was done using BUSCO [59]. Annotation was performed by using Blastp homology searches against the UniProt/SwissProt and UniProt/trEMBL databases [60]. Subsequently, gene expression heatmaps were constructed. A count table with expression values was used to produce heatmaps to visualise differential gene expression in http://www.heatmapper.ca [61]. The heatmaps were generated from normalised TPM (Transcripts Per Kilobase Million) of reads in the transcriptomes, generated from a specific tissue at a particular time. Genes with too low expression (< 5 TPM) were removed from the dataset. For the complete list of candidate genes investigated related to carpel, flower, fruit, dehiscence tissue development, and triacylglycerol, cuticle wax and cutin biosynthesis, see S4–S7 Tables.

**cDNA synthesis.** RNA was transcribed into cDNA through reverse transcriptase (RT) by using the iScript cDNA Synthesis Kit (Bio-RadLaboratories, Hercules, CA, USA) and 1 µg of extracted RNA. The reaction mix (20 µL) consisted of 1x iScript reaction mix, iScript reverse transcriptase (1 µL), 1 µg RNA template and water for the remainder of the reaction volume. Total cDNA was synthesized under the following conditions: priming of the RT at 25°C for 5 minutes; reverse transcription at 46°C for 20 minutes; RT inactivation at 95°C for 1 minute. Quantity and quality were measured with a DropSense Dropsense 96 (Trinean NV/SA, Gentbrugge, Belgium).

**Primer design.** A subset of 59 differentially expressed genes (*p-value* ≤ 0.05) of *E. pusilla* was selected to verify expression patterns using qPCR. The qPCR primers were designed based on *E. pusilla* transcriptome database (Orchidstra) annotations, and their identity of sequences are listed in S4–S7 Tables. We compared the transcriptome profile generated from RNA-seq analysis with those obtained from qPCR from fruit tissues of 1 WAP and 3 WAP. Nucleotide sequences of the reference genes *Actin* and *Ubiquitin* and selected differentially expressed genes of *E. pusilla* were downloaded from the transcriptome data. Specific primers were designed in OligoArchitect™ Online (Sigma-Aldrich®, Burlington, MA, USA) and Primer Quest™ (Integrated DNA Technologies, Inc., Coraville, Iowa, USA) (S8 Table). All primer pairs were blasted against the *E. pusilla* transcriptome database in Orchidstra [62] and further tested using PCR and gel electrophoresis. Primer sequences are listed in S8 Table and only exon-spanning primers with an amplification efficiency above 1.8 were used.

**Quantitative PCR (qPCR).** Full cDNA from three biological replicates of floral buds, and 1 WAP and 3 WAP *E. pusilla* fruits were analysed in biological triplicates for each primer set. For each amplicon group, a positive control was extracted from floral buds and fruits (from stages 1 to 14 WAP). A reaction without cDNA template was assigned as a negative control. The Luna Universal qPCR Master Mix M3003L (NEB Inc., Frankfurt am Main, Germany) was used according to the manufacturer's instructions, using 5 μM primer and 5 μl cDNA in a 1:50 dilution. The qPCR was run on a Lightcycler 480 II (Roche Diagnostics, Mannheim, Germany) in three biological replicates. *Actin* and *Ubiquitin* genes were used as reference genes and an interplate calibrator was used to adjust to interplate differences. The raw data was analyzed using the Pfaffl model [63] according to the MIQE guidelines [64].

## Results

### Ancestral state reconstruction

ASR is a method to evaluate the evolution of characters among species within a phylogenetic framework. The method aims to estimate character states back in time from observed states in extant species represented by the terminal nodes of a phylogenetic reconstruction. In this study, we were interested in the evolution of fruit traits in orchids and how they relate to each other, and carried out ASR on key characters present in the five orchid subfamilies (S3 Table). Firstly, an orchid phylogeny was calculated using nrITS, *mat*K and *rbc*L nucleic acid sequences (Fig 2). The main topology was congruent with the latest insights in the Tree of Life of orchids [33]. Ancestral state reconstructions were applied to analyse six fruit characters that were straightforward to score (Fig 2) and the Bayesian analyses results are summarized in Fig 3.

Individual results of the ASR analyses of six fruit characters are presented in Fig 2A–2F. Bootstrap values lower than 70% are indicated in Fig 2A with an asterisk. Although the major dichotomies were well-supported, it should be noted that lower values indicate that not all branches are well-supported, and the resulting low phylogenetic resolution might affect the interpretation of the evolution of a particular character state.

Throughout the phylogeny, a fruit ripening period shorter than four months was lost at least three times (Fig 2A), a pendant fruit orientation evolved at least twice from an erect orientation (Fig 2B), indehiscent fruits evolved from dehiscent fruits twice (Fig 2C), and the character less than three slits in mature fruits evolved at least three times from more than three slits (Fig 2D). Species sampling was too small to say much on the evolution of the character fused valves at the base only (Fig 2E). Remarkably, the character lignified DZ evolved multiple times from a non-lignified DZ (Fig 2F).

From the ASR analyses, out of 15 possible correlations, we discovered two correlations with strong support (LogBayes Factor 5 to 10), four with some support (LogBayes Factor > 2) and

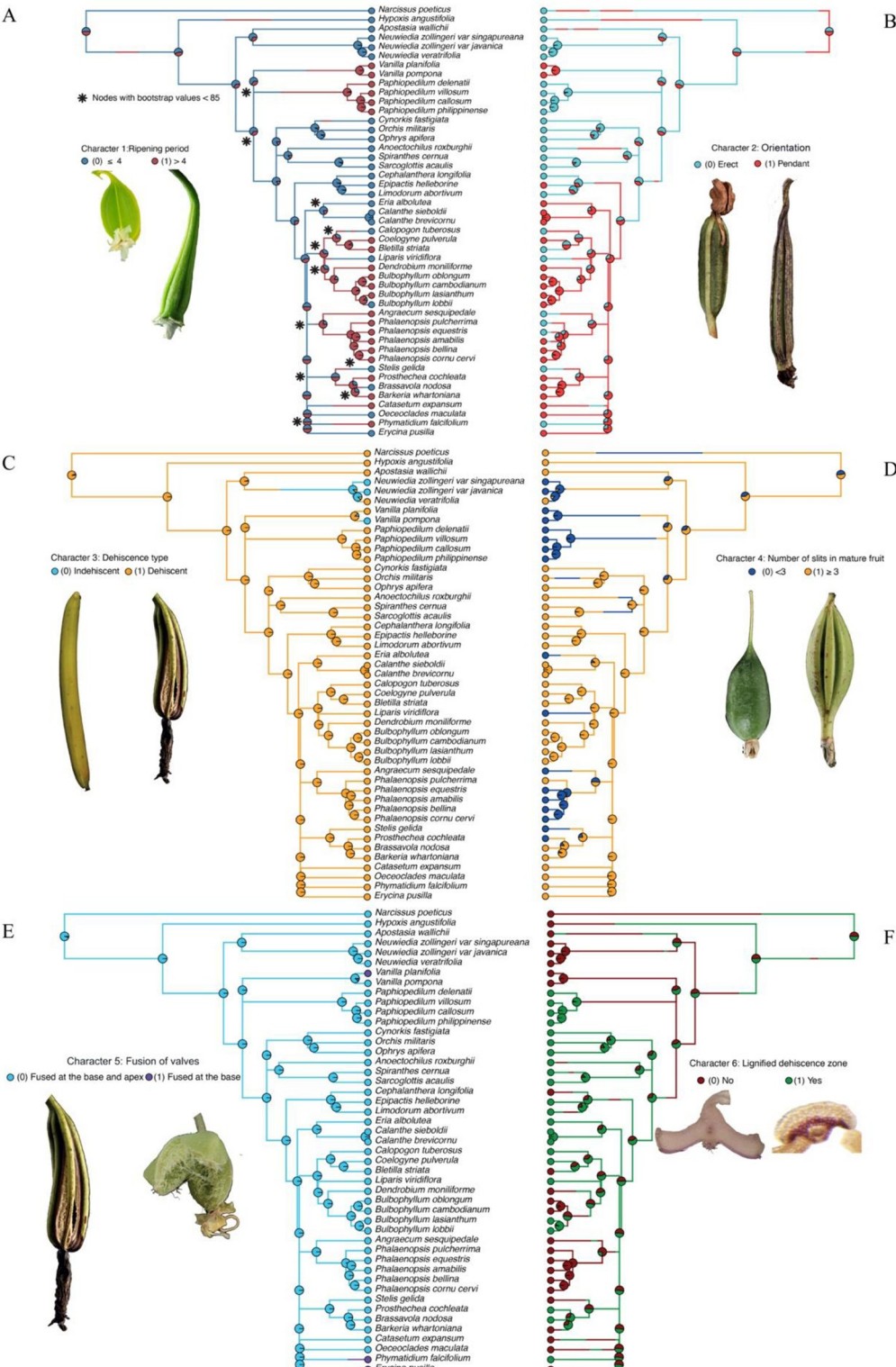

**Fig 2.** Ancestral state reconstruction of fruit ripening period (A), fruit orientation (B), fruit dehiscence type (C), number of slits in mature fruit (D), fusion of valves (E), and lignification of dehiscence zone (F) characters from stochastic mapping analysis based on joint sampling of nrITS, matK and rbcL (10,000 mapped trees). Posterior probabilities (pie charts) are mapped in a random stochastic character map. Note: * nodes have bootstrap values < 70%. Illustrations of fruits: (**A-B**) Character 1: *Erycina pusilla* (ripe in ≤4 months), *Phalaenopsis equestris*

(ripe in >4 months); character 2: *Habenaria eustachys* (erect), *Papilionanthe teres* (pendant). (**C-D**) Character 3: *Vanilla pompona* (indehiscent), *Arundina graminifolia* (dehiscent); character 4: *Lockhartia acuta* (<3 slits), *Guarianthe skinneri* (≥3 slits). (**E-F**) Character 5: *A. graminifolia* (valves fused at the base and apex), *Ornithocephalus valerioi* (valves fused at the base only); character 6: *E. pusilla* (non-lignified dehiscence zones), *Cynorkis fastigiata* (lignified dehiscence zones).

nine with weak support (LogBayes Factor < 2) (Fig 3). The traits ripening period and fruit orientation, on the one hand, and dehiscence type and number of slits in the mature fruit on the other hand, were found to be strongly correlated (LogBayes Factor > 5) (Fig 3).

To explore these correlations in more detail, we generated a transition map of ASRs of selected morphological characters from stochastic mapping analyses, which is depicted in Fig 4. There is evidence for the hypothesis that a short fruit ripening period (≤ 4 months) coevolved with a pendant fruit orientation (Fig 4A). The relation between the number of slits and fruit dehiscence type is ambiguous (Fig 4B). We found evidence of coevolution between fruits with valves fused at the base and apex with an erect fruit orientation (Fig 4C). Erect fruit orientation coevolved with an absence of lignification of the DZs (Fig 4D), and absence of lignification also coevolved with fruit dehiscence (Fig 4E). An orchid fruit with valves fused at the base coevolved with lignified DZs with a transition via fruits with valves fused at the base coevolving with non-lignified DZs (Fig 4F).

### Transcriptome sequencing of *E. pusilla* fruits

To obtain a comprehensive insight into transcriptional changes between 1 WAP and 3 WAP fruit stages, we carried out a transcriptome analysis. We found 83,137 gene loci, supported by RNA-seq, containing 35,245 (42%) coding loci with regular-sized proteins, and 47,892 putative

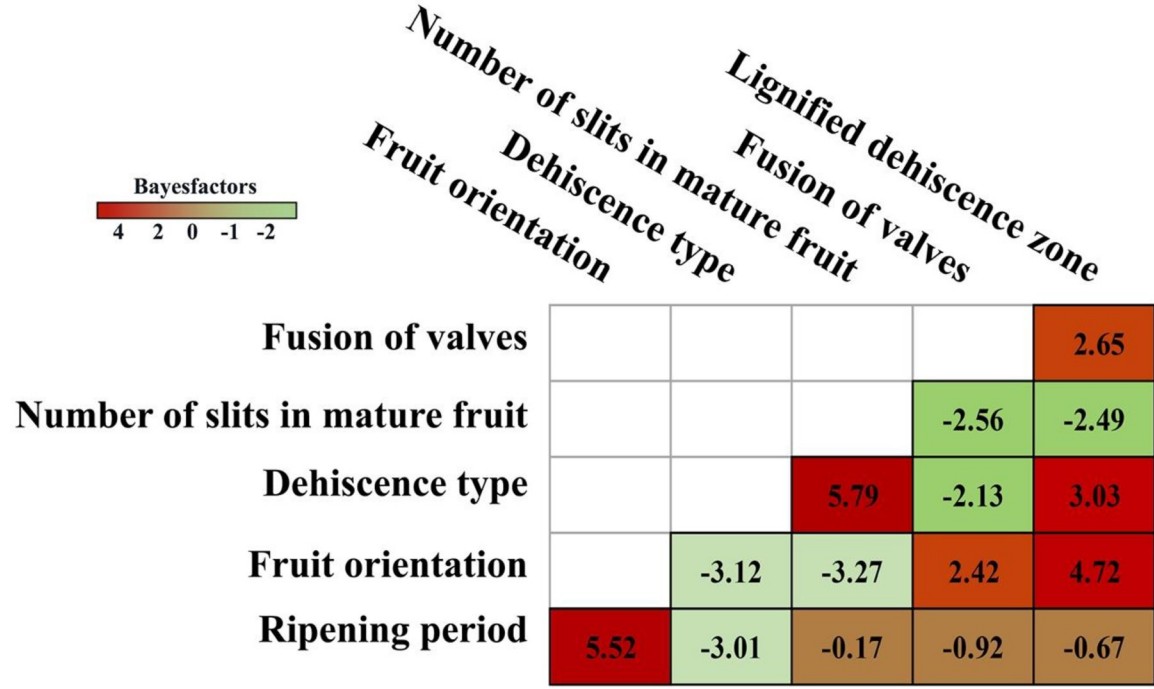

**Fig 3. Heatmap of correlations between each character analyzed for the ancestral state analyses.** Criteria: Bayesfactors > 10 very strong correlation, 5–10 strong correlation, >2 positive correlation, and <2 weak correlation.

A

**Character 2: Fruit orientation**
**[0] Erect [1] Pendant**
**Character 1: Ripening period (months)**
**[0] ≤ 4 [1] ≥ 4**

Erect and ripe ≤ 4 — 0.05 → Erect and ripe ≥ 4

← 5.09

14.16 | 0.49     14.23 | 13.79

Pendant and ripe ≤ 4 — 11.67 → Pendant and ripe ≥ 4

← 14.30

B

**Character 4: Number of slits in mature fruit**
**[0] < 3 [1] ≥ 3**
**Character 3: Dehiscence type**
**[0] Indehiscence [1] Dehiscence**

Slits < 3 and indehiscence — 5.24 → Slits < 3 and dehiscence

← 6.76

5.63 | 4.47     6.81 | 6.81

Slits ≥ 3 and indehiscence — 4.94 → Slits ≥ 3 and dehiscence

← 0.04

C

**Character 5: Fusion of valves**
**[0] Fused at the base and apex**
**[1] Fused at the base**
**Character 2: Fruit orientation**
**[0] Erect [1] Pendant**

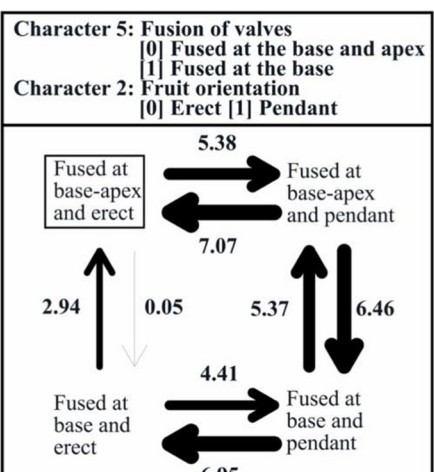

Fused at base-apex and erect — 5.38 → Fused at base-apex and pendant

← 7.07

2.94 | 0.05     5.37 | 6.46

Fused at base and erect — 4.41 → Fused at base and pendant

← 6.95

D

**Character 6: Lignified dehiscence zone**
**[0] No [1] Yes**
**Character 2: Fruit orientation**
**[0] Erect [1] Pendant**

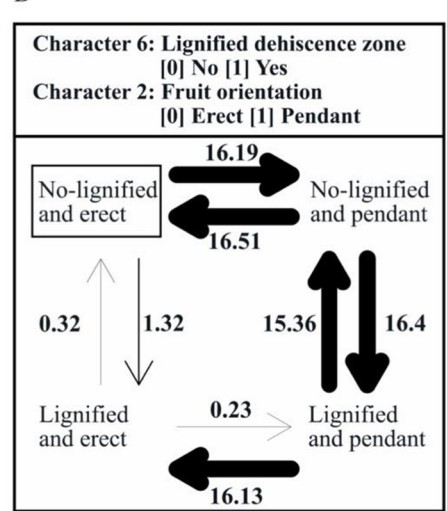

No-lignified and erect — 16.19 → No-lignified and pendant

← 16.51

0.32 | 1.32     15.36 | 16.4

Lignified and erect — 0.23 → Lignified and pendant

← 16.13

E

**Character 6: Lignified dehiscence zone**
**[0] No [1] Yes**
**Character 3: Dehiscence type**
**[0] Indehiscence [1] Dehiscence**

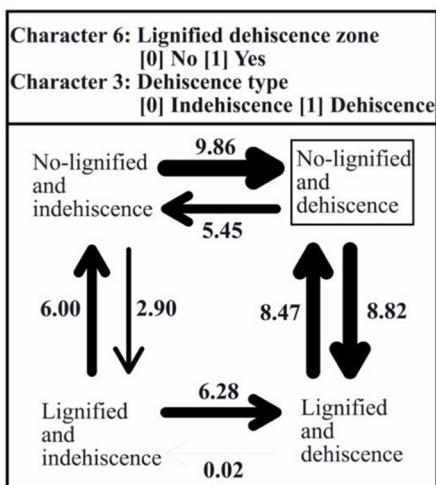

No-lignified and indehiscence — 9.86 → No-lignified and dehiscence

← 5.45

6.00 | 2.90     8.47 | 8.82

Lignified and indehiscence — 6.28 → Lignified and dehiscence

← 0.02

F

**Character 5: Fusion of valves**
**[0] Fused at the base and apex**
**[1] Fused at the base**
**Character 6: Lignified dehiscence zone**
**[0] No [1] Yes**

Fused at base-apex and no-lignified — 4.44 → Fused at base-apex and lignified

← 7.06

6.63 | 7.18     2.03 | 0.13

Fused at base and no-lignified — 7.35 → Fused at base and lignified

← 3.43

**Fig 4.** Ancestral state reconstructions of (A) character 2 (fruit orientation) vs 1 (ripening period), (B) character 4 (number of slits in mature fruit) vs 3 (dehiscence type, (C) character 5 (fusion of valves) vs character 2 (fruit orientation), (D) character 6 (lignified dehiscence zone) vs character 2 (fruit orientation), (E) character 4 (number of slits in mature fruit) vs character 3 (dehiscence type), and (F) character 5 (fusion of valves) vs character 6 (lignified dehiscence zone) from stochastic mapping analyses based on a joint sampling of nrITS, *mat*K and *rbc*L (10000 map trees). The numbers represent the estimated number of evolutionary changes and the time spent in each state, arrows represent the direction of the transition between the states, and character combinations indicated with a square are interpreted as having co-evolved.

loci with small proteins (smORF < 120 aa). A total of 218,753 alternate transcripts were found at 35,676 (43%) loci. A total of 55,054 (66%) of 83,137 coding genes could be translated into complete proteins, and 28,083 into partial proteins. The assembled unigenes were annotated by using Blastp homology searches against the UniProt/SwissProt and UniProt/trEMBL databases, resulting in 16,987 (6.38%) reliable annotations, of which 54,290 (20.39%) were shorter than the reference proteins, 4,182 (1.57%) were longer than the reference proteins, 97,772 (36.73%) had only partial similarity to a reference, 17,379 (6.53%) matched a putative transposon, and for 75,604 (28.40%) no homology was found.

Gene expression was calculated as transcripts per million (TPM). With a cut-off TPM $\geq$ 5, we calculated the number of transcripts identified in 1 WAP and 3 WAP tissues of *E. pusilla* fruits. With this criterion, we found 40,677 transcripts with TPM counts $\geq$ 5. The number of transcripts was highest in the earlier fruit stage (1 WAP) compared to the number of transcripts in the later stage (3 WAP) with 21,864 and 18,813 transcripts, respectively (Fig 5).

## Identification of differentially expressed genes (DEGs) by RNA-seq

Putative differentially expressed genes (DEGs) were identified using the following criteria: (1) *p-value* $\leq$ 0.05 and (2) fold change (FC) $\geq$ 2 for upregulated and $\leq$ –2 for down-regulated DEGs. With these criteria, a total of 2,568 upregulated and 3,217 downregulated reliable transcripts were identified in 1 WAP and 3 WAP of *E. pusilla* fruits (Fig 6).

Expression of putative dehiscence regulatory genes in the tissues investigated ranged from entirely absent (e.g., *EpMADS24*/$B_{sister}$ in 1 WAP fruits) to approximately 989.82–1176.90 TPM (e.g. *EpMALD3* in 1 WAP fruits). We selected genes with a cut-off TPM $\geq$ 5. By generating heatmaps, we studied the expression profile of a wide range of genes, including MIKC

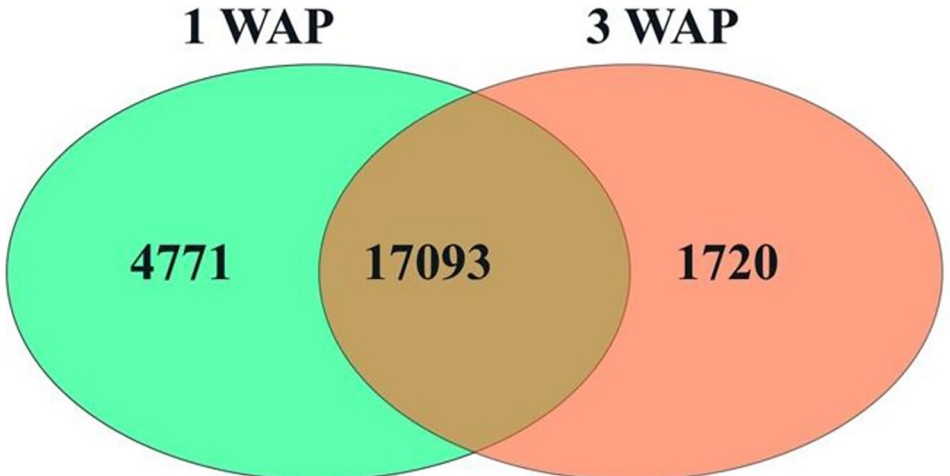

**Fig 5. Venn diagrams of identified genes (cutoff TPM $\geq$ 5) in 1 WAP and 3 WAP fruits of *Erycina pusilla*.**

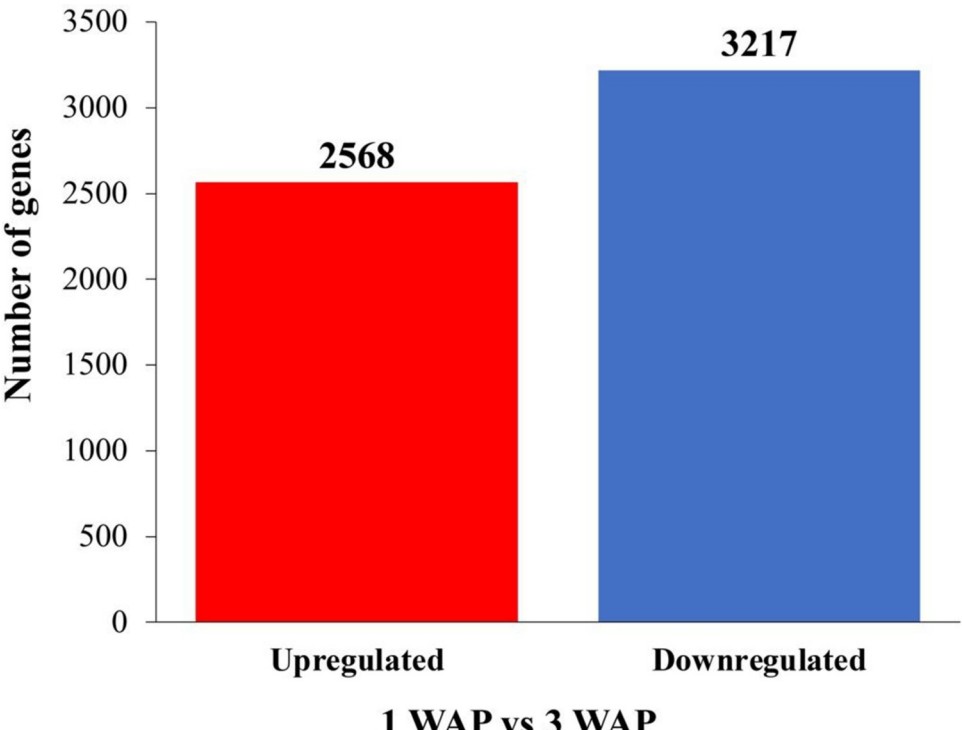

**Fig 6. Changes in gene expression profile of 1 WAP and 3 WAP fruits of *Erycina pusilla*.** Upregulated genes are shown in red, while the number of downregulated genes is shown in blue.

MADS-box transcription factors (S2 Fig), fruit dehiscence development genes (S3 Fig), carpel, gynoecium, ovule, and fruit developmental genes (S4 Fig), and triacylglycerols, cuticle wax and cutin biosynthesis genes (S5 Fig). Finally, we carried out a differential expression analysis of each gene expressed in each developmental stage tested and continued with an analysis of variance (ANOVA) (IBM SPSS statistics version 26) (S4–S7 Tables). The main results are summarized below.

**Expression of MIKC MADS-box genes in *E. pusilla* fruits.** MIKC MADS-box transcription factors (TFs) have pivotal roles in determining floral organ and fruit identity. In the *E. pusilla* transcriptome, many MADS-box TFs were found to be expressed in 1 WAP and 3 WAP fruits. Some were expressed evenly throughout fruit development, such as *EpSEP2/3* and *EpAGL6-1* (S4 Table; S2 Fig).

In contrast, *EpSEP1–4* showed substantial differences in expression dynamics. *EpSEP4* was hardly expressed whereas *EpSEP2* were equally expressed in all tissues investigated. *EpAG1*, *EpAG1, AG2*, *EpAG3*, *EpAP3-1/2*, and *EpSEP1* were significantly expressed in 1 WAP fruits. *EpSOC1*, a floral integrator gene [65, 66], and *EpB_{sister} (EpMADS24/EpBS)*, a *MADS-box* gene reported to be associated with orchid ovule integument development [66, 67], were significantly expressed in 3 WAP fruits (S4 Table; S2 Fig).

qPCR was used to validate the expression pattern of 18 MADS-box genes using qPCR. We discovered discrepancies between the TPM from RNA-seq and the expression pattern from qPCR results in some MADS-box genes. Only three MADS-box genes showed similar trends between RNA-seq and qPCR with a strong positive correlation ($r \geq 0.5$ and *p-value $\leq 0.05$*; S10 Table.

**Expression of fruit development and dehiscence genes in *E. pusilla* fruits.** We also found transcripts of many orthologs of genes related to DZ, valve and replum formation in *A.*

*thaliana* in 1 WAP and 3 WAP *E. pusilla* fruits. These genes are related to the degradation of the middle lamella during fruit dehiscence in *A. thaliana* [10]. These effectors include fiber biosynthesis genes, lipid biosynthesis and their interaction with plant hormones such as auxin, gibberellic acid, cytokinin, etc. For example, a cellulose gene encoding for cellulose cell wall loosening *(CEL5)* [68] and pectinase gene *(PG1)* [69, 70]. *PG1* expression in the DZ is regulated by *IND* and *HEC3* transcription factors [71]. Other genes associated with the development of the DZ in *A. thaliana* are the cellulase genes *KORRIGAN 1/KOR1* [72], *CEL6*, and the hemicellulase gene *MAN7* [73]. In addition, several genes are involved in the auxin pathway of *A. thaliana* carpel, gynoecium, and ovule development such as *ARF3/ETTIN* [74], and *ARF6* [75, 76]; *TPL* [77–79]; *CLHC1* [80]; and *PIN1B* [81].

The following orthologs did not show differential expression: *EpKNAT2, EpKOR1, MAN7, EpCEL5, EpYAB2, EpARF6, EpTPL, EpPIN3a, EpCLHC1, EpAP2-10,* and *EpAP2-11* (S5 Table; S3 Fig).

*EpRPL, EpHEC1* (the homolog of *AtHEC1*, a gene that controls carpel fusion and gynoecium development in *A. thaliana* [82]), *EpPG1, EpARF15,* and *EpPIN1B* were higher expressed in 1 WAP fruits as compared to 3 WAP fruits. The opposite was found for *EpHEC1, EpIAA17, EpARF3* which were higher expressed in 3 WAP fruits as compared to 1 WAP fruits (S5 Table; S3 Fig).

The correlation between RNA-seq and RT-qPCR results demonstrated that only three out of twenty fruit development and dehiscence genes have a strong positive correlation $r \geq 0.5$ and *p-value* $\leq 0.05;$ S10 Table.

**Expression of carpel, gynoecium, and ovule developmental genes.**   We also found transcripts of many orthologs of genes related to carpel, gynoecium and ovule development in *A. thaliana* in 1 WAP and 3 WAP *E. pusilla* fruits. For example the *ARR1/RR21, ARR12/RR23* and *CKX3* genes are related to the cytokinin pathway. They are expressed during gynoecium development in *A. thaliana*, particularly in meristematic tissues [83, 84]. *RGA* encodes *DELLA* proteins that negatively regulate gibberellic acid responses during stamen and petal development in *A. thaliana* [85].

The following genes did not show differential expression: *EpCRC/DL1, EpSLR1, EpMSG2* [86], *EpCKX3, EpARR1/RR21,* and *EpBEE1* [76]. In addition, *EpCIN2, EpCIN3, EpPCF2* and *EpPCF3*, orthologs of the *TCP* gene family, involved in multiple aspects of plant growth [87–89], showed an equal expression across different fruit developmental stages.

Several transcription regulators (TRs) have essential functions in carpel, gynoecium and ovule development of *A. thaliana* and *P. equestris*, e.g. *LUG, SEU* and *LUH* [90–95], *CYP40/ SQN* [76], *PCF1* [96], and *PCF8* [96]. The following orthologs genes were higher expressed in 1 WAP compared to 3 WAP fruits: *EpMYB305, EpLUG, EpSQN, EpPCF1,* and *EpULT1*. In contrast, *EpPCF8* and *EpLUH* were higher expressed in 3 WAP compared to 1 WAP fruits (S6 Table; S4 Fig).

We discovered differences in the TPM of RNA-seq and qPCR results. Only two carpel and ovule development genes showed similar trends and a strong positive correlation between RNA-seq and qPCR analyses out of the six tested genes $r \geq 0.5$ and *p-value* $\leq 0.05;$ S10 Table.

**Expression of triacylglycerol, cuticle wax and cutin biosynthesis genes.**   Lipids occur in plants in the extracellular domain as cuticles and waxes, as storages in the form of triacylglycerol, and as membranes [97]. Genes related to TAG biosynthesis include the acetyl-CoA synthesis *E. pusilla* orthologs of *ACP1* [98–100], *BCCP1* [98], fatty acid synthesis genes *KCS11* [98, 101], *MOD1* [98, 102, 103] and *KASI2* [98, 102, 104], fatty acid modified genes *2FAD7* [98, 102, 105] and *SLD1* [98, 102], and TAG assembly genes *LPAT5* and *GPDHp* [102]. Genes associated with wax biosynthesis are *WSD1* [106], alkaline formation genes *CER1, CER3* and Cyt b5 [107], the cutin polyester synthesis gene *GPAT6* [108], and wax transporter genes *ABCG11*,

*ABCG12*, *ABCG13* and *BDG1* [109–112]. In cutin biosynthesis of *A. thaliana*, several genes are associated with acyl chains oxidase, acyltransferase, intercellular midchain hydroxylase, cutin polyester transport, and incorporation of hydroxyacyl monomers into polymers [98, 108, 110, 112–115]. We analyzed transcripts related to triacylglycerol, cuticle wax and cutin biosynthesis in the fruits of *E. pusilla* and found putative orthologs of 66 *A. thaliana* fatty acid biosynthesis genes (S7 Table; S5 Fig).

In addition to the list of TAG, wax, and cutin biosynthesis genes, several genes associated with wax biosynthesis in *A. thaliana* were found, including *AS1* [18, 116], *CAC3* [102], *ESR2* [109], *PDG6* [117], *MIXTA* [118, 119], *nsLTPs* [120, 121], and *MALD3* [109].

The following genes did not show differential expression: *EpLEC2*, of which the ortholog in *A. thaliana* is a TF regulating seed maturation [122, 123], *EpACP1*, *EpACLA2*, *EpACP3*, *EpACP4*, *EpKAS12*, *EpFAD7*, *EpFAD12*, *EpSLD1*, *EpGPDH*, *EpCER10*, *EpKCR1*, *EpCER3*, *EpCYP77A4*, *EpGATP6*, *EpGATP9*, *EpABCG11*, *EpABCG12*, *EpABCG3*, *EpE1-BETA-2*, *EpMIXTA*, *EpEREBP1*, *EpLPEAT1*, *EpMALD3*, *EpRAP2-4*, *EpRAP2-6*, *EpRAP2-12*, *EpSF3B2* and *EpDCR*.

The following genes were found to be differentially expressed among 1 WAP and 3 WAP fruits: *EpFUS3*, *EpBCCP1*, *EpACLA3*, *EpEMB3147*, *EPKCS11*, *EpKCS2*, *EpMOD1*, *EpFAR3*, and *EpLACS1*, of which the ortholog in *A. thaliana* is involved in transformation of fatty acids [124], *EpPAS2* of which the ortholog in *A. thaliana* encodes for fatty acyl-CoA reductase [125] were higher expressed in 1 WAP fruits whereas *EpKCS20*, *EpGL1*, *EpLPAT5*, *EpLACS4*, *EpCER6*, *EpCER26-like*, *EpCYP86A8*, *EpMAH1* were higher expressed in 3 WAP fruits.

Only four of 16 triacylglycerol, cuticle wax and cutin biosynthesis genes biosynthesis genes tested exhibited a similar pattern, with a strong positive correlation between RNA-seq and qPCR ($r \geq 0.5$ and *p-value ≤ 0.05;* S10 Table.

## Validation of subset of differentially expressed genes by quantitative real-time PCR (qPCR)

Several genes were identified as being potentially involved in *E. pusilla* fruit development by DEG analysis following RNA-seq. To verify patterns observed we applied qPCR.

A total of 59 DEGs were selected for validation. Of these 59 genes, 20 were MADS-box genes related to fruit development (*EpFUL1*, *EpFUL2*, *EpFUL3*, *EpPI*, *EpAG1*, *EpAG2*, *EpAG3*, *EpAGL6-1*, *EpAGL6-2*, *EpAGL6-3*, *EpSEP1*, *EpSEP2*, *EpSEP3*, *EpSEP4*, *EpMADS18/SVP*, *EpFVF*, *EpMADS24/B$_{sister}$*, *EpAP2-3*, and *EpAP2-10*), five were involved in carpel and gynoecium development (*EpCUC1*, *EpCUC2*, *EpCUC3*, *EpDL1*, and *EpSLR1*), one was involved in ovule development (*EpPCF2*), 13 were related to fruit dehiscence (*EpIND/SPT*, *EpHEC3*, *EpPG1*, *Ep KNAT1*, *EpKNAT2*, *EpKOR1*, *EpCEL5*, *EpMAN7*, *EpARF6*, *EpBLH9*, *EpIAA17*, *EpERF10*, and *EpTPL*), ten regulated triacylglycerol biosynthesis (*EpFUS3*, *EpA-CL1.3A*, *EpACP4*, *EpKCS2*, *EpKCS6*, *EpMOD1*, *EpPDH-E1 BETA*, *EpFAD7*, *EpSLD1*, and *EpGL1*), seven were associated with wax biosynthesis (*EpPAS2*, *EpCER3*, *EpMAH1*, *EpAS1*, *EpESR/EREBP1*, *EpMIXTA*, and *EpPDG6*), and three were linked to cutin biosynthesis (*EpABCG11*, *EpABCG12*, and *EpBDG1*) (S4–S7 Tables).

We analyzed the qPCR data with one-way ANOVA to identify DEGs. Among 59 genes validated with qPCR, we found 21 DEGs (S8 and S9 Tables; Fig 7). Among those genes, we found four MADS-box genes (*EpAG2*, *EpAGL6-3*, *EpSEP3*, and *EpMADS24/B$_{sister}$*), three fruit development and dehiscence genes (*EpTPL*, *EpRPL*, and *EpKOR1*), two carpel and ovule development genes (*EpDL1*, and *EpSLR1*) and four lipid biosynthesis genes (*EpKCS6*, *EpGL1*, *EpPHD-E1*, and *EpMAH1*) that had similar trends between RNA-seq and qPCR analyses ($r \geq 0.5$ and *p-value ≤ 0.05;* S10 Table).

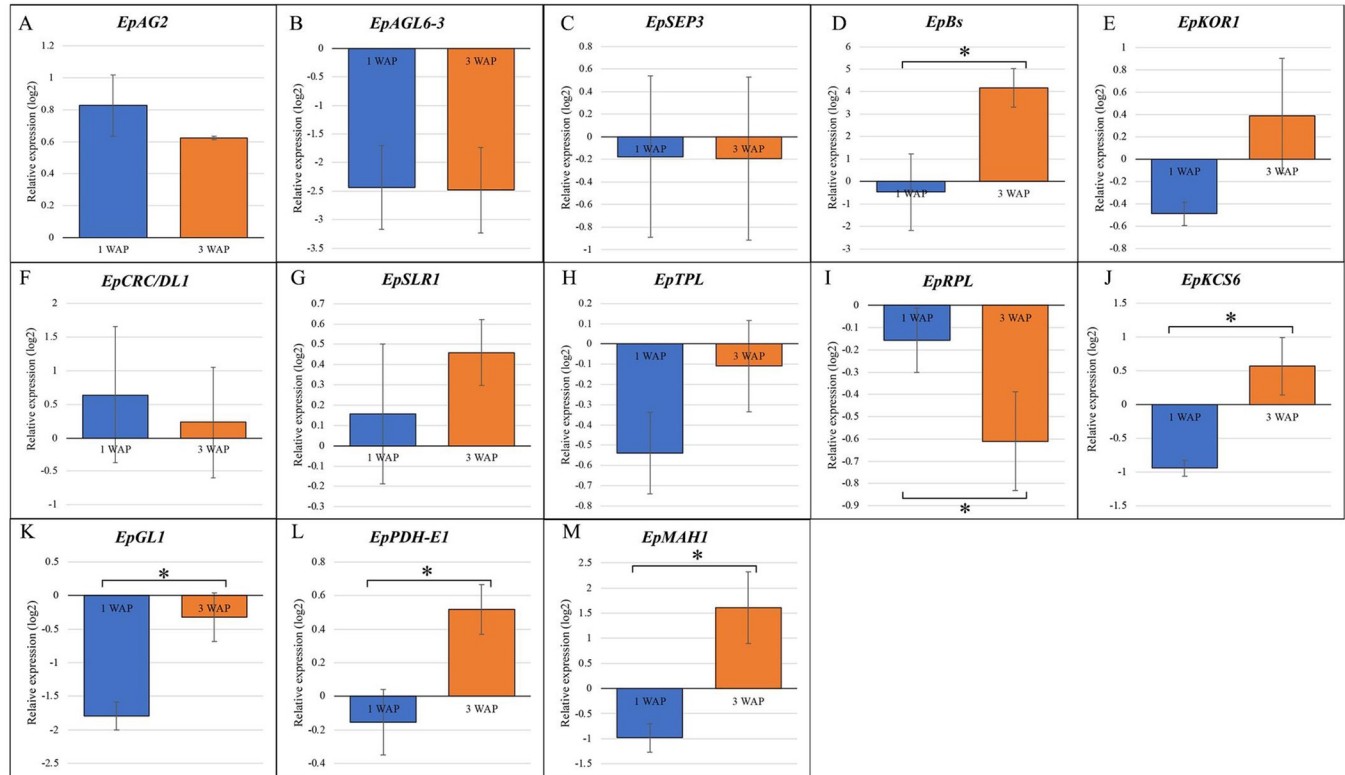

**Fig 7.** Relative expression of MADS-box (A-D), fruit development and ovule, and carpel development (E-I), and lipid biosynthesis (J-M) genes from qPCR analysis in one week after pollination (1 WAP) and three weeks after pollination (3 WAP) fruit valves of *E. pusilla*. The x-axis shows the two samples of 1 WAP and 3 WAP. The y-axis represents the relative expression (in log2) of genes as the mean of three biological replicates' expression and normalized relative to floral bud tissues. The asterisks denote statistically significant differences in the relative expression of the various developmental stages (*$p \leq 0.05$).

## Discussion

### Ancestral character state analysis of orchid fruits

The species studied represent a great diversity in fruit characteristics, growth habits, and phylogenetic position within the orchid family. We were able to sample four species from both genera of Apostasioideae (one of *Apostasia* and three species of *Neuwiedia*), two species from *Vanilla* as representative of the 15 genera in Vanilloideae, four species from *Paphiopedilum* as representative of the five genera in the Cypripedioideae, six species from six out of 650 Orchidoideae genera (one species of *Anoectochilus, Cynorkis, Ophrys, Orchis, Sarcoglottis,* and *Spiranthes*), and 29 species from 21 genera (one species of *Angraecum, Barkeria, Bletilla, Brassavola, Calopogon, Catasetum, Cephalanthera, Coelogyne, Dendrobium, Epipactis, Eria, Erycina, Limodorum, Liparis, Oeceoclades, Phymatidium, Prosthechea,* and *Stelis;* two species of *Calanthe;* four species of *Bulbophyllum;* and five species of *Phalaenopsis*) representing the more than 575 Epidendroideae genera (S1 Table). Despite the fact that the study covered representatives of all subfamilies, some fruit characters are still underrepresented in our sampling, such as the characters dehiscence type and fusion of fruit valves. Indehiscence was only represented by *Vanilla pompona, Neuwiedia zollingeri* and *Neuwiedia zollingeri*. Likewise, for the fusion of fruit valves, we only found basally fused valves in three species: *Vanilla planifolia, Erycina pusilla,* and *Phymatidium falcifolium*. Increasing the sample size to include more species for these two fruit characters could provide us with more unambiguous evidence and information about evolutionary changes in these characters.

We found that fruit ripening period ($\leq 4$ or $> 4$ months), fruit orientation (erect or pendant), reinforcement of DZs (lignified or non-lignified) and fruit opening (dehiscent or indehiscent) are phylogenetically informative within the orchid family. But numerous reversals also indicate some plasticity of these characters during orchid evolution. Plasticity of fruit characters is also reported for Brassicaceae [126–129]. Most Brassicaceae have dehiscent fruits, like orchids, but some genera have indehiscent fruits such as *Cardaria* Desv [130]. Occurrence of plastic fruit characters might be explained by adaptation to different ecological niches (e.g. temperature, season) and/or dispersers [131]. For example, in *Aethionema* indehiscent fruits are an adaptation to arid areas where rainfall is irregular [126]. Under these circumstances, delaying seed dehiscence maintains seed germination capacity [126]. Whether this is also the case for indehiscent orchid fruits needs further investigation by comparing ecological conditions and seed dispersal agents of dehiscent orchid species like *E. pusilla* with indehiscent orchid species like *Vanilla pompona*. In general, more species and ecological characters related to fruit development should be included in future studies.

Our study showed that an erect fruit orientation is the most ancestral character in orchids. In a previous analysis of the inflorescence orientation in *Phalaenopsis*, a pendant inflorescence also evolved from an erect one [132]. Previously it was reported that in a subalpine plant community, erect fruits manage to disperse their seeds over much greater distances and have a more even distribution of dispersal direction than pendant ones [133]. Whether this also holds for erect orchid inflorescences and fruits is still unknown.

According to our analyses, an erect orchid fruit co-evolved with a more extended ripening period and non-lignified DZs. Longer fruit ripening periods are mostly found in tropical orchids that generally grow in humid areas with less distinct seasons, allowing for an extension of fruit and seed developmental time. Fruits with a lignified DZ were found in temperate orchids or tropical orchids with a short growing season, in which fruits and seeds must ripen relatively quickly. In non-lignified DZs of orchids, upon fruit maturation, lignification only occurs in a single layer of the endocarp and vascular bundles, particularly in sclerenchyma fiber caps (see S1F and S1H Fig). No sclerenchymatous cells were found in other pericarp layers. The DZ of *Phalaenopsis equestris* fruits consists of two layers of small cells, similar to *E. pusilla* (S1G and S1H Fig) and *Oncidium flexuosum* [31, 134, 135]. Along the DZs, instead of lignification a lipid layer is formed. Future study of lipid content by liquid chromatography-mass spectrometry (LC-MS) may shed more light on the quantity and types of lipids in these DZ tissues.

### Are there genes similarly expressed during dehiscence zone development in *A. thaliana* and *E. pusilla* fruits?

The *A. thaliana* fruit DZ consists of both a lignified and unlignified separation layer [10]. *AtFUL*, which belongs to the *AP1/FUL* clade, and *AtSHP1/2*, which belongs to the *AG* clade, are both involved in the formation of lignified DZs in *A. thaliana*. *SHP1* and *SHP2* are the primary genetic regulators of DZ development in *A. thaliana* [11]. The conserved function of *SHP* in fruit lignin deposition has been reported for *Prunus persica* and *Nicotiana benthamiana* [136–138]. Several MADS-box genes with sequences and expression patterns similar to *SHP* have been identified in orchids. In line with protein-protein interactions found in *A. thaliana*, [31] reported distinct expressions of the *AG* homologs *EpMADS21* and *EpMADS22* and interaction with *FUL* homologs *EpMADS11* and *EpMADS12*. Yet, it is still unclear whether *AtSHP* and *EpAG* have the same function in the specification of DZ formation in fruits. Assuming *EpMADS21* and *EpMADS22* are *AtSHP* homologs, the difference in expression pattern of these genes compared with *AtSHP1/AtSHP2* expression may be due to gene

duplication and subfunctionalization of *AG* in the *E. pusilla* genome. Functional changes and genetic differentiation may have allowed morphological changes in the dehiscence zone of *E. pusilla* compared to *A. thaliana*. Further characterization and functional analyses of *SHP* lineages from various plant species is needed to further investigate this hypothesis.

Previously it was reported that *EpMADS11/FUL2* interacts with *EpMADS21/AG1-EpMADS22/AG2* and *EpSEP2/SEP4* to regulate fruit development in *E. pusilla* [31]. Our RNA-seq and qPCR results show that all *FUL*-clades were lowly expressed in both 1 WAP and 3 WAP fruits of *E. pusilla*. In contrast, *EpMADS22/AG2* and *EpSEP3* were found to be equally high expressed in 1 WAP and 3 WAP fruits (S9 Table). These results indicate that the interaction between *EpAG2 -EpSEP3* in the absence of *EpFUL*-copies might partly regulate valve and DZ formation in orchid fruits. Yeast two-hybrid experiments using domain proteins of *AG*, *FUL*, and *SEP* need to be conducted in order to understand protein-protein interactions. Functional characterization of MADS-box motifs also needs to be carried out to confirm whether these genes play a role in the formation of DZs in the *E. pusilla* fruit.

*EpMADS24/EpBs* was lower expressed in 1 WAP as compared to 3 WAP fruits (Fig 7). The involvement of this gene in orchid fruit valves or DZ development has never been reported before, indicating possible novel function of this gene in regulating development of those tissues.

The *A. thaliana* DZ is located between the valves and the replum of the fruits. *FUL* limits dehiscence gene expression to the valve margin. *FUL* functions in the valve, while *RPL* functions in the replum but is not required for replum development [12, 13, 17, 139]. *RPL* was significantly higher expressed in 1 WAP than 3 WAP *E. pusilla* fruits, implying that *RPL* may be involved in early valve and possibly also DZ formation.

Specification of the development of the lignified and non-lignified separation layers of the valve margin requires expression of *IND* [14], whereas *ALC* and its paralog, *SPT* are involved in the formation of the separation layer in *A. thaliana* [15]. According to the qPCR results, *EpIND/HEC3* and *EpALC/SPT* were lowly expressed and did not show significant expression patterns in either 1 or 3 WAP fruits (S9 Table). In orchids, a non-lignified DZ co-evolved with a dehiscent fruit. The absence of expression of *IND/HEC3* and *ALC/SPT* in the fruit valve of *E. pusilla* might be correlated with the absence of lignified DZ tissue. Low expression of these genes is also found in *Lepidium appelianum*, a Brassicaceae species with indehiscent fruits [136]. Interestingly, a change in *IND/ALC* expression in *Lepidium* caused the loss of DZ formation and fusion of the endocarp layer next to the lignified vascular bundle, leading to indehiscent fruits [122]. Decrease in expression of *IND/HEC3* and *ALC/SPT* in *E. pusilla* may correlated with non-lignified DZs but not with fruit indehiscence.

Processes downstream of fruit dehiscence formation involve cell wall rupture, particularly the degradation of the middle lamella [140, 141]. The cellulose gene *KOR1* plays a role in cell wall degradation in *A. thaliana*. In *E. pusilla*, *KOR1* was evenly expressed in 1 WAP and 3 WAP fruits. This gene is involved in cell wall degradation and needs to be expressed during maturation of dehiscent fruits. Our results indicate that expression of this gene may only become upregulated at orchid fruit maturation stage (> 11 WAP) (Figs 7 and 8, S9 Table).

Validating the expression pattern of genes found to be significantly expressed in *E. pusilla* fruits requires functional analysis. Inactivation of genes in conjunction with protein-protein interactions studies can assist in identifying genes that regulate DZ formation in the *E. pusilla* fruit. In addition, whole-transcriptional analysis of orchids with lignified DZs is necessary to reveal differences between the regulatory networks of lignified and non-lignified DZs. Co-expression and interaction analysis can also be performed using an *in silico* approach to complete information regarding the network of genes interacting during formation of lignified versus non-lignified DZs in orchid fruits.

*Erycina pusilla* fruit
One week after pollination

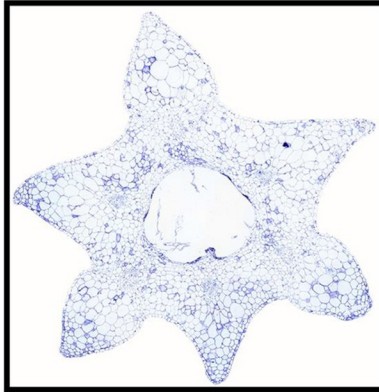

*Erycina pusilla* fruit
Three weeks after pollination

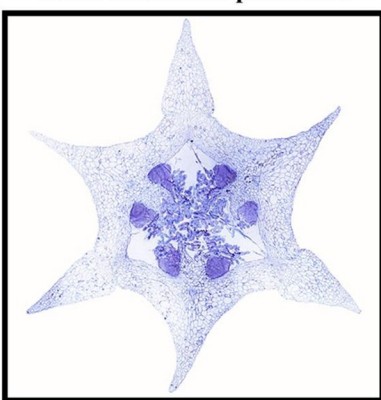

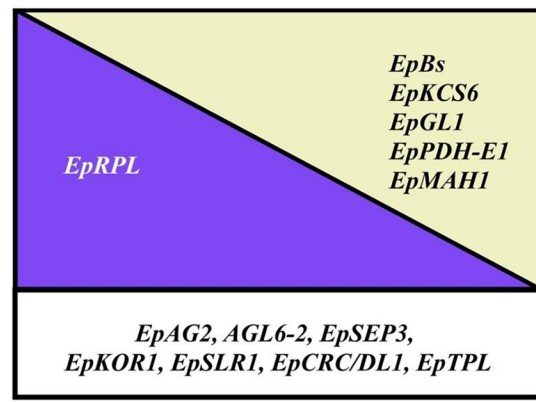

**Fig 8. Summary of expression patterns assessed by RNA-seq and qPCR of genes involved in *Erycina pusilla* fruit development.** Expression of *EpRPL* gene is higher (*indicated in the purple triangle*) in 1 WAP fruits but lower in 3 WAP fruits. Expression of *EpBs*, *EpKCS6*, *EpGL1*, *EpPDH-E1*, and *EpMAH1* genes is higher (*indicated in the yellow triangle*) in 3 WAP fruits (*right side*) but lower in the 1 WAP fruits. Expression of *EpAG2*, *EpAGL6-2*, *EpSEP3*, *EpKOR1*, *EpSLR1*, *EpCRC/DL1*, and *EpTPL* genes is high (*indicated by the white box*) in the 1 WAP and 3 WAP fruits.

## Carpel and gynostemium genes are expressed during dehiscence zone development of *E. pusilla* fruits

Two carpel and gynostemium genes were equally expressed in 1 and 3 WAP *E. pusilla* fruits. The first one was *EpDL1*, an ortholog of the *CRC* gene that belongs to the *YAB* transcription factor encoding gene family, with both activator and repressor functions in *A. thaliana*. It acts as an activator of gene transcription required for carpel fusion, septum formation, and nectary development [142]. In *Oryza sativa*, expression of *DL*, a *CRC* ortholog, was constrained to developing floral meristem, carpels, and leaf midrib [143]. In orchids, *DL* homologs seem to have acquired novel regulatory functions following their duplication. *EpDL1*, was upregulated in 1 and 3 WAP fruits, indicating involvement in the formation of orchid fruit valves and possibly also DZ formation (Figs 7 and 8). In *P. equestris*, *PeDL1* and *PeDL2* are involved in ovule and gynostemium development [144]. *PeDL2*, which interacts with B-class genes (*PeMADS 2–5*), was also found to be involved in perianth development [145].

The second carpel and gynostemium gene found to be expressed in 1 and 3 WAP *E. pusilla* fruits was DELLA protein *SLEDER RICE 1 (SLR1)*, an ortholog of *AtGAI/AtRGA2*. *OsSLR1* that is negatively regulating gibberellic acid signaling [146]. In orchids, the *SLR1* gene was previously found to be expressed during flower development of *Arundina graminiflolia* [147] and in seeds of *Dendrobium catenatum* [148]. *SLR1* gene expression in both 1 WAP and 3 WAP fruits of *E. pusilla* suggest a possible additional function of this gene in regulating orchid fruit development.

## Triacylglycerol, cuticle wax and cutin biosynthesis genes are expressed during dehiscence zone development in *E. pusilla* fruits

Many triacylglycerol, cutin and wax biosynthesis genes were upregulated in our qPCR data. However, most of these genes were expressed evenly in both 1 WAP and 3 WAP tissue of *E. pusilla* fruits. An exception were *EpKCS6*, *EpGL1*, *EpPDH-E1*, and *EpMAH*. All of these genes were higher expressed in 3 WAP compared to 1 WAP fruits (Figs 7 and 8, S9 Table). *PDH-E1* (*E1-BETA-2 Pyruvate dehydrogenase E1*) is a gene related to fatty acid synthesis. Ke [149] reported this gene to be involved in silique development and fatty acids, needed in early fruit

development of *A. thaliana*. We presume that the activity of fatty acid synthase genes in *E. pusilla* fruit valves is related to the production of lipid content to support early fruit valve growth. The *AtKCS6* (*3-ketoacyl-CoA synthase 6*) gene is known to play an important role in the production of cuticular wax [150]. Very long fatty acids (VLCFAs) are precursors for the synthesis of waxes in pollen husks, fruits, stems and leaves [151]. The *MAH1* gene is involved in the formation of wax alcohols and ketones in the wax biosynthetic pathway in *A. thaliana* [152]. Its homolog, *EpMAH1*, was significantly upregulated in 3 WAP fruits (Figs 7 and 8). Whether or not this gene is involved in the formation of the lipid layer present in the DZ tissue of *E. pusilla* fruits requires further research. Future research may employ a combination of virus-induced gene silencing (VIGS) and in situ hybridization to determine function and localization of *EpKCS6*, *EpPDH-E1*, *EpGL1*, and *EpMAH1* expression in DZ tissue of 3 WAP fruits.

According to the ASR that we carried out, a lignified DZ in orchid fruits evolved from a non-lignified one. It might be that the metabolic pathway to lignin in plants is derived from the one to lipids. Indications for this were found by [153] who detected that the lipid-phenolic matrix of the moss species *Physcomitrella patens* reveals extant representatives of a common ancestor of the suberin, cutin and lignin polymers that differentiate vegetative tissues in more recent plant lineages. We found upregulated expression of *EpKCS6*, *EpPDH-E1*, and *EpMAH1* in the 3 WAP *E. pusilla* fruits, but low expression of genes associated with the formation of lignified DZs such as *IND/HEC3* and *ALC/SPT*. Further investigations into the amount and type of lignin and lipids in DZs of different orchid species, combined with additional gene expression analysis, will shed more light on the evolutionary transition from non-lignified to lignified DZs in orchid fruits.

## Conclusions

We conclude that erect dehiscent fruits with non-lignified DZs and a short ripening period are ancestral characters in orchids. Lignified DZs in orchid fruits evolved multiple times from non-lignified DZs. MADS-box and other genes involved in carpel and gynoecium development, and lipid, wax and cutin biosynthesis in *A. thaliana* show different expression patterns in *E. pusilla* fruits or are absent from the genome. Interestingly, the homologs of *MADS24/B$_{sister}$*, lipid biosynthesis genes *KCS6*, *GL1*, *PHD-E1*, and *MAH1* were highly expressed in 3 WAP *E. pusilla* fruits, indicating a possible novel function of these genes in the regulation of *E. pusilla* fruit development and dehiscence. Our findings indicate that the current *A. thaliana* fruit dehiscence model should be adapted for orchids.

## Supporting information

**S1 Fig.** Cross-sections of developing *E. pusilla* (A-F) and *P. equestris* fruits (G-H), embedded in LR White and stained with toluidine blue. (**A**) Fruit 7 days after pollination (DAP). (**B**) Magnified part of the sterile valve at 2 weeks after pollination (WAP). Arrows indicate the dehiscence zone. Black boxes the exo-, meso- and endocarp layer. (**C**) Fruit of 2 WAP. (**D**) Fruit of 4 WAP. (**E**) Fruit of 16 WAP. (**F**) Magnified part of the sterile valve at 16 WAP. (**G**) Fruit of 120 DAP. (**G**) Magnified part of the sterile valve at 120 DAP; cell with thick blue stained wall part of vascular bundle (VB) indicates sclerenchyma fiber cap. Black arrows indicate the dehiscence zone. F, fertile valve; S, sterile valve; PT, pollen tube. Scale bar (C, K) = 0.2 mm, (D) = 0.1 mm, (E–I) = 1 mm, (J, G, H) = 0.5 mm. [Figs A-F from [31]; Figs G-H captured by Dewi Pramanik].
(DOCX)

**S2 Fig. RNA-seq heatmap (TPM) of MICK-MADS-box transcription factors transcriptionally active in one week after pollination (1 WAP) and three weeks after pollination (3 WAP) fruits of *E. pusilla*.**
(DOCX)

**S3 Fig. RNA-seq heatmap (TPM) of fruit dehiscence developmental genes transcriptionally active in one week after pollination (1 WAP) and three weeks after pollination (3 WAP) fruits of *E. pusilla*.**
(DOCX)

**S4 Fig. RNA-seq heatmap (TPM) of carpel, gynoecium, and ovule developmental genes transcriptionally active in one week after pollination (1 WAP) and three weeks after pollination (3 WAP) fruits of *E. pusilla*.**
(DOCX)

**S5 Fig. RNA-seq heatmap (TPM) of triacylglycerols, cuticle wax, and cutin biosynthesis genes transcriptionally active in one week after pollination (1 WAP) and three weeks after pollination (3 WAP) fruits of *E. pusilla*.**
(DOCX)

**S1 Table. Orchid fruit traits scoring.**
(XLSX)

**S2 Table. Characters and character states defined for ancestral state reconstructions.**
(XLSX)

**S3 Table. Accession number of *rbc*L, *mat*K, nrITS sequences of orchid species used for ancestral state reconstruction analyses.**
(XLSX)

**S4 Table. Transcript counts in TPM of selected genes MADS-box genes.**
(XLSX)

**S5 Table. Transcript counts in TPM of selected genes related to fruit development.**
(XLSX)

**S6 Table. Transcript counts in TPM of selected genes related to carpel and gynoecium development.**
(XLSX)

**S7 Table. Transcript counts in TPM of selected genes related with triacylglycerol/TAG biosynthesis.**
(XLSX)

**S8 Table. Primer design for qPCR.**
(XLSX)

**S9 Table. Summary of ANOVA test of qPCR results.**
(XLSX)

**S10 Table. Pearson bivariate correlation analysis of RNA-seq (log10) versus qPCR (log10) results.**
(XLSX)

## Acknowledgments

Andrea Weisert from the Botanical Institute of Justus Liebig University is thanked for her assistance with the qPCRs.

## Author Contributions

**Conceptualization:** Dewi Pramanik, Annette Becker, Anita Dirks-Mulder, Erik Smets, Barbara Gravendeel.

**Data curation:** Dewi Pramanik, Clemens Roessner, Oliver Rupp, Anita Dirks-Mulder, Kevin Droppert, Alexander Kocyan.

**Formal analysis:** Dewi Pramanik, Clemens Roessner, Oliver Rupp, Diego Bogarín, Oscar Alejandro Pérez-Escobar, Anita Dirks-Mulder, Alexander Kocyan.

**Funding acquisition:** Dewi Pramanik.

**Investigation:** Dewi Pramanik, Anita Dirks-Mulder, Kevin Droppert.

**Methodology:** Dewi Pramanik, Annette Becker, Diego Bogarín, Anita Dirks-Mulder, Barbara Gravendeel.

**Supervision:** Barbara Gravendeel.

**Validation:** Dewi Pramanik, Annette Becker, Kevin Droppert.

**Visualization:** Dewi Pramanik, Clemens Roessner, Diego Bogarín, Anita Dirks-Mulder, Kevin Droppert.

**Writing – original draft:** Dewi Pramanik, Erik Smets, Barbara Gravendeel.

**Writing – review & editing:** Dewi Pramanik, Annette Becker, Clemens Roessner, Erik Smets, Barbara Gravendeel.

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
