## [Decision Letter · Decision Letter 0]

4 Apr 2023

PONE-D-23-02491Evolution and development of orchid fruitsPLOS ONE

Dear Dr. Pramanik,

Thank you for submitting your manuscript to PLOS ONE. After careful consideration, we feel that it has merit but does not fully meet PLOS ONE’s publication criteria as it currently stands. Therefore, we invite you to submit a revised version of the manuscript that addresses the minor points raised during the review process.

We look forward to receiving your revised manuscript.

Kind regards,

Serena Aceto, Ph.D.

Academic Editor

PLOS ONE

“This study was financially supported by a personal grant from Badan Penelitian dan Pengembangan Pertanian, Indonesia (SMARTD-IAARD) to Dewi Pramanik.”

Reviewers' comments:

Reviewer's Responses to Questions

**Comments to the Author**

1. Is the manuscript technically sound, and do the data support the conclusions?

Reviewer #1: Partly

Reviewer #2: Yes

2. Has the statistical analysis been performed appropriately and rigorously? 

Reviewer #1: Yes

Reviewer #2: Yes

3. Have the authors made all data underlying the findings in their manuscript fully available?

Reviewer #1: Yes

Reviewer #2: Yes

4. Is the manuscript presented in an intelligible fashion and written in standard English?

Reviewer #1: Yes

Reviewer #2: Yes

5. Review Comments to the Author

Reviewer #1: The title seems too wide since only one species Erycina pusilla in addition to other data obtained from the

Genebank were adopted if I am right. All the analyses seems reasonable by using standard bioinformatic tools, with only concern about the treatment of plant materials. You grew the plants in in vitro condition, however, the conditions on pollination, subsequent plant and capsule growth, and capsule harvesting were not well described. I guess you pulled out the plants from the flask, did the pollination and regrown in the flask until capsule sampling. Or if you did the sampling of the capsule in greenhouse grown plants, then what is the reason doing in vitro cultivation since this orchid is so tiny? One more thing is in the abstract, you used abbreviation E. pusilla for the orchid then later used full genus name. It is suggested to use full genus name when first appeared.

Reviewer #2: Reviewer’s Comments:

Comments for the Author:

Pramanik et al. performed ancestral state reconstructions across the five orchid subfamilies to study the evolution of selected fruit traits and explored dehiscence zone developmental genes by RNA-seq and qPCR, which We found that erect dehiscent fruits with non-lignified dehiscence zones and a short ripening period are ancestral characters in orchids. Furthermore, they carried out gene expression analysis of tissues from different developmental stages of E. pusilla fruits, found that fruit dehiscence genes were regulated by the MADS-box gene family and other important regulators. Finally, they discovered that homologs of A. thaliana genes involved in the development of carpel, gynoecium and ovules, and genes involved in lipid biosynthesis were expressed in the fruit valves of E. pusilla. It has promoted the research of the evolution and development in orchid fruits.

However, there are a few points that need improvement:

1. At the beginning, the summary “statereconstructions” should be split into two words: “state reconstructions”.

2. The title of the article should be added to the first page of the main text.

3. Please strictly follow the journal format when citing references in the main text, as this may be confused with the markings on lines 179-183.

4. The background section is too wordy, please focus on the background of fruit development and orchid plants.

5. Figure 2 is too blurry to see clearly.

6. In this paper, you only studied E. pusilla, which may not represent the overall fruit development of orchid plants. Appropriate modifications need to be made in the conclusion section on lines 787 and 790.

6. PLOS authors have the option to publish the peer review history of their article (what does this mean?). If published, this will include your full peer review and any attached files.

Reviewer #1: No

Reviewer #2: No

---

## [Author Response · Author response to Decision Letter 0]

9 May 2023

> Editor comments: 

1) Please ensure that your manuscript meets PLOS ONE’s style requirements, including those for file naming. The PLOS ONE style templates can be found at

Response: Thank you for your comment. As requested, we adjusted the manuscript according to the journal’s style requirements. 

2) Thank you for stating the following financial disclosure:“This study was financially supported by a personal grant from Badan Penelitian dan Pengembangan Pertanian, Indonesia (SMARTD-IAARD) to Dewi Pramanik.” Please state what role the funders took in the study. If the funders had no role, please state: “The funders had no role in study design, data collection and analysis, decision to publish, or preparation of the manuscript.” Please include this amended Role of Funder statement in your cover letter; we will change the online submission form on your behalf. If this statement is not correct, you must amend it as needed. 

Response: We rewrote the funding role: “This study was financially supported by a personal grant from the Sustainable Management for Agriculture Research and Development (SMARTD) project of Badan Penelitian dan Pengembangan Pertanian, Indonesia number 133/KPTS/Kp.320/02/2018 to Dewi Pramanik. The funders had no role in study design, data collection and analysis, publication decision, or manuscript preparation”. We have also included information about the role of funding in our review cover letter. 

Response: Thank you for your comment. We ensured that the title in the submission form (via Edit Submission) is identical to the one in the revised manuscript.

Response: Thank you for your suggestion. We emphasize the results of differential expressions rather than only the TPM values. Therefore, we deleted the sentence “with TPM values ranging from 0 to 57,487 (data not shown)”.

5. Please review your reference list to ensure it is complete and correct. If you have cited papers that have been retracted, please include the rationale for doing so in the manuscript text, or remove these references and replace them with relevant current references. Any changes to the reference list should be mentioned in the rebuttal letter that accompanies your revised manuscript. If you need to cite a retracted article, indicate the article’s retracted status in the References list and also include a citation and full reference for the retraction notice.

Response: We have checked and updated every reference and made sure that the revised reference list is complete and correct. 

> Reviewer 1 comments: 

>The title seems too wide since only one species Erycina pusilla. 

Response: We agree with your suggestion that the title of the original manuscript was too broad. We therefore changed it into: "Evolution and development of fruits of Erycina pusilla and other orchid species". We also changed sub titles in results and discussion sections accordingly.

> in addition to other data obtained from the Genebank were adopted if I am right. All the analyses seems reasonable by using standard bioinformatic tools, with only concern about the treatment of plant materials. You grew the plants in in vitro condition, however, the conditions on pollination, subsequent plant and capsule growth, and capsule harvesting were not well described. I guess you pulled out the plants from the flask, did the pollination and regrown in the flask until capsule sampling. Or if you did the sampling of the capsule in greenhouse grown plants, then what is the reason doing in vitro cultivation since this orchid is so tiny? 

Response: The full life cycle of E. pusilla was completed under in vitro conditions, from seed harvesting, seed germination, plantlet maturation, flower pollination, fruit production, to fruit harvesting. Detailed information on the tissue culture and pollination of E. pusilla is provided in “Molecular analysis” section, lines 307-325.

> One more thing is in the abstract, you used abbreviation E. pusilla for the orchid then later used full genus name. It is suggested to use full genus name when first appeared.

Response: Thank you for your detailed review of our manuscript. We changed E. pusilla into Erycina pusilla when it first appears in the abstract.

> Reviewer 2 comments:

>Pramanik et al. performed ancestral state reconstructions across the five orchid subfamilies to study the evolution of selected fruit traits and explored dehiscence zone developmental genes by RNA-seq and qPCR, which We found that erect dehiscent fruits with non-lignified dehiscence zones and a short ripening period are ancestral characters in orchids. Furthermore, they carried out gene expression analysis of tissues from different developmental stages of E. pusilla fruits, found that fruit dehiscence genes were regulated by the MADS-box gene family and other important regulators. Finally, they discovered that homologs of A. thaliana genes involved in the development of carpel, gynoecium and ovules, and genes involved in lipid biosynthesis were expressed in the fruit valves of E. pusilla. It has promoted the research of the evolution and development in orchid fruits.

Response: Thank you very much for your kind words. 

>However, there are a few points that need improvement:

1. At the beginning, the summary “statereconstructions” should be split into two words: “state reconstructions”.

Response: We corrected the sentence according to your suggestion.

>2. The title of the article should be added to the first page of the main text.

Response: We added the title to the first page of the revised manuscript.

>3. Please strictly follow the journal format when citing references in the main text, as this may be confused with the markings on lines 179-183.

Response: We revised the references following the journal’s Vancouver (PLOS ONE) reference style. To avoid confusion, we placed each citation after each statement instead of putting all citations at the end of the sentence.

>4. The background section is too wordy, please focus on the background of fruit development and orchid plants.

Response: Thank you for your suggestion. We removed irrelevant information on fruit development and orchids from the introduction. We considered information on genes related to fruit development, dehiscence, carpel, gynoecium, ovule development and lipid biosynthesis from Arabidopsis thaliana relevant as a reference to compare results of gene expression analyses of orchid fruits with, but moved this information from the introduction to the results section as indicated below: 

1. Information on genes related to development and dehiscence of fruits was moved to the Results section, lines 561-570.

2. Information on genes related to carpel, gynoecium, and ovule development was moved to the Results section, lines: 587-591.

3. Information on genes related to triacylglycerol, cuticle wax and cutin biosynthesis was moved to the Results section, lines: 610-619.

>5. Figure 2 is too blurry to see clearly.

Response: We have uploaded improved figures with better resolution

>6. In this paper, you only studied E. pusilla, which may not represent the overall fruit development of orchid plants. Appropriate modifications need to be made in the conclusion section on lines 787 and 790.

Response: This statement was revised according to your suggestion.

---

## [Decision Letter · Decision Letter 1]

24 May 2023

Evolution and development of fruits of Erycina pusilla and other orchid species

PONE-D-23-02491R1

Dear Dr. Pramanik,

We’re pleased to inform you that your manuscript has been judged scientifically suitable for publication and will be formally accepted for publication once it meets all outstanding technical requirements.

Kind regards,

Serena Aceto, Ph.D.

Academic Editor

PLOS ONE

Additional Editor Comments (optional):

Reviewers' comments:

Reviewer's Responses to Questions

**Comments to the Author**

1. If the authors have adequately addressed your comments raised in a previous round of review and you feel that this manuscript is now acceptable for publication, you may indicate that here to bypass the “Comments to the Author” section, enter your conflict of interest statement in the “Confidential to Editor” section, and submit your "Accept" recommendation.

Reviewer #2: All comments have been addressed

Reviewer #3: (No Response)

2. Is the manuscript technically sound, and do the data support the conclusions?

Reviewer #2: Yes

Reviewer #3: Yes

3. Has the statistical analysis been performed appropriately and rigorously? 

Reviewer #2: Yes

Reviewer #3: Yes

4. Have the authors made all data underlying the findings in their manuscript fully available?

Reviewer #2: Yes

Reviewer #3: Yes

5. Is the manuscript presented in an intelligible fashion and written in standard English?

Reviewer #2: Yes

Reviewer #3: Yes

6. Review Comments to the Author

Reviewer #2: The authors have answered our questions or suggestions well, so we considered the manuscript to be accepted and published in this journal.

Reviewer #3: (No Response)

7. PLOS authors have the option to publish the peer review history of their article (what does this mean?). If published, this will include your full peer review and any attached files.

Reviewer #2: No

Reviewer #3: **Yes: **Jian-Zhi huang

---

## [Editor Report · Acceptance letter]

30 May 2023

PONE-D-23-02491R1 

Evolution and development of fruits of *Erycina pusilla* and other orchid species 

Dear Dr. Pramanik:

I'm pleased to inform you that your manuscript has been deemed suitable for publication in PLOS ONE. Congratulations! Your manuscript is now with our production department. 

Kind regards, 

on behalf of

Dr Serena Aceto 

Academic Editor

PLOS ONE